# How Families Manage the Complex Medical Needs of Their Children with *MECP2* Duplication Syndrome

**DOI:** 10.3390/children10071202

**Published:** 2023-07-11

**Authors:** Dani John Cherian, Daniel Ta, Jeremy Smith, Jenny Downs, Helen Leonard

**Affiliations:** 1School of Human Sciences, University of Western Australia, Perth 6009, Australia; dani.johncherian@telethonkids.org.au (D.J.C.); jeremy.smith@uwa.edu.au (J.S.); 2Telethon Kids Institute, Centre for Child Health Research, University of Western Australia, Perth 6872, Australia; daniel.ta@telethonkids.org.au (D.T.); jenny.downs@telethonkids.org.au (J.D.); 3School of Medicine, University of Western Australia, Perth 6009, Australia; 4Curtin School of Allied Health, Curtin University, Perth 6845, Australia

**Keywords:** *MECP2* duplication syndrome, neurodevelopmental disorder, intellectual disability, epilepsy, recurrent infections, multimorbidity, complex care, caregiver wellbeing, rare diseases

## Abstract

*MECP2* duplication syndrome (MDS) is a rare, X-linked, neurodevelopmental disorder resulting from the duplication of the methyl-CpG-binding protein 2 (*MECP2*) gene. The clinical features of MDS include severe intellectual disability, global developmental delay, seizures, recurrent respiratory infections, and gastrointestinal problems. The aim of this qualitative study was to explore how the parents of children with MDS manage their child’s seizures, recurrent respiratory infections, and gastrointestinal symptoms, and the impact on them as parents. The data were coded into three categories: (1) complex care needs in the home, (2) highly skilled caregivers, and (3) impact on caregivers and families. Complex 24 h care was required and parents developed complex skillsets to ensure that this was delivered well to their child. The provision of extensive complex medical care in the home had an impact on parent mental and physical health, family dynamics, and finances. This study captures the management of high-burden comorbidities in MDS at home. Investigations into how best to support caregiver wellbeing to reduce their stresses, whilst maintaining optimal child health and wellbeing, are needed.

## 1. Introduction

In 1999, Lubs and colleagues discovered an X-linked syndrome that caused severe central nervous deterioration and multiple respiratory infections [1]. More than 20 years later, this disorder is now known as *MECP2* duplication syndrome (MDS; OMIM 300260), a rare neurodevelopmental disorder caused by the duplication of the methyl-CpG-binding protein 2 (*MECP2*) gene. The clinical features of MDS include, but are not limited to, infantile hypotonia, global developmental delay, severe intellectual disability, regressive motor skills, recurrent respiratory infections, epilepsy, and gastrointestinal problems [2,3,4,5,6]. In addition, affected individuals may have behavioural characteristics such as attentional difficulties, hand stereotypies, hyperactivity, bruxism, uncontrolled screaming spells, and autistic features [7,8].

Despite its very low birth-prevalence of 1/150,000 liveborn males, knowledge about MDS is gradually accumulating [2]. A salient milestone was the development of the international *MECP*2 Duplication Database (MDBase) in 2020, which aims to provide a clearer international understanding of the clinical features and natural history of individuals with MDS [5]. Prior to and subsequent to this publication there have been a limited number of publications with a sample size sufficiently adequate to characterise the aspects of this disorder [3,4,5,9,10,11,12,13,14]. Despite this, there is growing information about the prevalence and time course of the medical comorbidities affecting individuals with MDS [5]. For example, over 60% of individuals with MDS had seizures and the median age of onset was 8 years for males and 9.8 years for females. More than 60% of individuals with MDS were also reported to have experienced pneumonia from birth to the age of 20 years [5].

### 1.1. Respiratory Health and Infections

Poor respiratory health is mainly characterised by recurrent respiratory infections [12]. Data from MDBase identified that over 80% of individuals with MDS (118/146) had been hospitalised multiple times due to a respiratory problem [5]. Other respiratory symptoms such as breathlessness, obstructive sleep apnoea, nocturnal hyperventilation, bronchospasms, and chronic coughing/wheezing may be a consequence of bronchiectasis and reactive airway disease [8]. Swallowing difficulties may cause the aspiration of food and fluids, leading to the development of aspiration pneumonia [13]. The inability to clear chest secretions and change body position due to hypotonia may further contribute to the risk of respiratory infections. Manual airway clearance techniques such as postural drainage and chest physiotherapy can assist in the management of excess respiratory secretions. Other methods for airway clearance include suctioning, vest- and cough-assisted therapy, and positive expiratory pressure (PEP) masks [8].

Individuals with MDS may be more susceptible to infections with encapsulated bacteria such as *Haemophilus influenzae* and *Streptococcus pneumoniae* due to a weak immune response towards the polysaccharide capsule of these bacteria [13]. Immunodeficiency combined with a weak cough contributes to multiple respiratory tract infections, particularly in the winter months. As a result, short- and long-term courses of antibiotics are often recommended, along with attention to vaccinations and boosters [8,13].

### 1.2. Seizures

Over half (n = 326) of 619 individuals from 76 case studies were reported to experience refractory seizures by the age of 9 years [8,15]. Furthermore, a large 2022 cross-sectional study (n = 101) found that caregivers perceived epilepsy as being the most prominent symptom of MDS [9]. The types of seizures such as tonic, clonic, absence, and gelastic seizures may vary between individuals, with many having multiple seizure types, as seen in those diagnosed with Lennox–Gastaut syndrome [4]. Epilepsy treatment options for individuals with MDS can include single or polytherapy regimens of anti-seizure medications or medicinal cannabis. Surgical procedures such as corpus callosotomy or vagal nerve stimulation are occasionally implemented [5,15]. Currently, no monotherapy or polytherapy regime has been shown to effectively control seizures for this disorder.

### 1.3. Gastrointestinal Health

Individuals with MDS often have poor gastrointestinal health. The most common gastrointestinal problem encountered is chronic constipation. A recent study (n = 106) characterising gastrointestinal health in MDS revealed that symptoms commonly associated with constipation occurred in over 70% of individuals with MDS [10]. Other gastrointestinal symptoms included abdominal bloating, gastroesophageal reflux, slow gut motility, air swallowing, and intestinal pseudo-obstruction [3,5,12]. Strategies for improving gastrointestinal health may include the management of constipation and enteral feeding.

### 1.4. Individuals with MDS Require Complex Care

Overall, there are limited treatments that substantially reduce the symptoms of MDS. As a result, there is a critical need for future research regarding the treatment of the clinical symptoms of MDS to improve the patient’s overall health and quality of life. Furthermore, individuals with MDS require complex care. This comprises specialist support for chronic health conditions and assistance in managing day-to-day symptoms and daily activities at home [16]. Providing complex medical care in the home can be extremely stressful and the burden can place significant emotional, physical, and financial burdens on families and caregivers [17,18,19,20]. Due to the severity of disability and multimorbidity, the medical management of MDS is expected to be challenging and intensive; however, there is currently no research which investigates the family management of the complex care required for MDS.

### 1.5. Coordination of Complex Care

The parents of children with complex health needs are faced with a significant responsibility of coordinating care for their child, oftentimes with minimal external support depending on the health and welfare system of the country in which they live [19,20,21,22]. For example, the period of transition to home following a hospital admission can be a stressful time for parents [23]. Outside of the hospital setting, parents may be solely responsible for organising, prioritising, and attending all the various medical appointments required for their child, with some families attending two to four appointments a week [19]. Furthermore, they can also be faced with the challenge of sourcing special medical equipment and medications which can be particularly difficult if the family is located in a rural setting [24,25]. Moreover, resourcing skilled carers who are able to provide complex care at home can be challenging due to the variation in health systems for different countries as well as the family’s economic resources. This challenge is amplified if the external carer suddenly leaves or is no longer available due to reductions in healthcare packages [23,26].

### 1.6. Performing Medical Procedures and Managing Emergencies

The parents of children with complex health needs have to perform medical procedures at home as part of their child’s complex care and learning how to perform these tasks with limited medical training can be challenging and emotionally taxing [18]. Furthermore, parents may also experience anxiety regarding the operation of medical devices for their child at home.

A qualitative study conducted in 2020 exploring complex care at home for children with tracheostomies found that some parents experienced constant anxiety regarding the functioning of their child’s ventilation machines, even with a carer present [19]. Lastly, managing emergencies for children with complex disabilities can be extremely stressful for parents, which is further amplified if the child is non-verbal. This may leave parents to interpret and assess whether or not their child is having a medical emergency [27,28]. Having such a state of constant anxiety for these situations can have a detrimental effect on the parents’ mental health and overall wellbeing [17].

### 1.7. Effect of Complex Care on the Caregiver’s Physical Health

The parents of children requiring complex care spend a large amount of time ensuring the best possible health outcome for their child, leaving them with minimal time to manage their own physical health [29,30]. Some caregivers may experience a lack of sleep as they provide intensive complex care for their child during the day, followed by various medical routines at night such as giving feeds and medications and responding to equipment alarms [31,32,33,34]. The emotional demands of providing complex care at home may also be a contributing factor in caregivers experiencing sleep difficulties [35]. Furthermore, the lack of sleep can also result in suboptimal care for the children, as some caregivers may find it difficult to perform certain medical procedures correctly when they are fatigued [19].

### 1.8. Effect of Complex Care on the Caregiver’s Mental Health and Relationships

The intensity of medical routines for children and young adults with severe disabilities can significantly impact the parent’s mental health. The emotional impact of the daily home-based medical care can leave caregivers with feelings of anger, anxiety, guilt, and helplessness [17]. A large 2022 cross-sectional study investigating the burden on caregivers of MDS through a quantitative scale found that higher burden scores were directly correlated to increased levels of anxiety, depression, and emotional exhaustion [11]. Furthermore, intensive caregiving can significantly impact family relationships and dynamics. For example, devoting large amounts of time and energy to one child requiring complex care can cause feelings of guilt for parents with multiple children as they may feel that they are providing inadequate attention to the other children [19,36]. Furthermore, the siblings of a child with complex medical needs may also experience feelings of trauma as they witness their brother/sister’s clinical symptoms and medical procedures at home [37,38,39].

Home-based medical management can also significantly impact a family’s social life, including challenges in going out, which can be due to practical constraints such as carrying large medical equipment around or the perceived risk of infection outside of the home [40]. Caregivers may also be worried about leaving their child in the care of another relative or friend as they may not have the proper training to provide the complex care required [19,41].

Despite a growing understanding of the medical comorbidities of individuals with MDS, no study has explored how parents and caregivers medically manage their child’s complex symptoms at home. This study aimed to explore how families manage their child’s complex health needs, particularly their management of epilepsy, recurrent respiratory infections, and gastrointestinal symptoms, and the impact on them as parents and caregivers. This study also aimed to investigate what families needed to provide complex care well and the strategies that supported their needs. We believe that caregiver perspective can inform education and awareness about MDS for healthcare professionals and possibly contribute to the development of future clinical care guidelines.

## 2. Materials and Methods

### 2.1. Study Design and Ethics

This was a qualitative descriptive study. Ethical approval was granted by the UWA Human Research Ethics Committee (RA/4/20/5929). At the commencement of the interviews, all participants gave verbal informed consent, which was recorded and documented in the interview transcripts.

### 2.2. Participants

We sought to understand experiences of providing complex medical care in the home in an international sample of parents with a child of any age with MDS. Purposive sampling across clinical characteristics was chosen because of the limited literature in the field. Participant selection aimed for maximal variation to document different experiences across children’s ages and the management of epilepsy, respiratory, and/or gastrointestinal problems. Participants were recruited from the international *MECP*2 Duplication Database (MDBase) [42]. Parent caregivers were invited by telephone to participate in the study. Upon acceptance of invitation, an interview date was arranged, and participants were provided with an information sheet and a copy of the interview schedule by email.

### 2.3. Procedures

Interview data were collected to generate personal narratives describing the provision of their child’s complex care at home. The interview schedule (Appendix A) comprised open-ended and probing questions about the child’s health needs and management protocols of their epilepsy, gastrointestinal, and respiratory symptoms. More specifically, the day-to-day routines of daily medical care such as medication delivery, administrative challenges including coordination of care, and problem solving methods for resolving adverse health events were also explored. The interview schedule also included questions exploring the impact on the parent and family, and strategies that helped or hindered. Training was provided by senior researchers (J.D., H.L.) to the primary researcher (D.J.C.).

Interviews were conducted using videoconference facilities and were audio-recorded and transcribed verbatim. Some audio recordings of the interviews were transcribed manually by D.J.C. and others were transcribed professionally using transcription software. Interview transcripts were emailed to participants for their review and editing if they wished to do so.

### 2.4. Data Analysis

Thematic analysis of the reviewed interview transcripts was informed by the framework method approach which identifies commonalities and variations in qualitative data to draw descriptive and explanatory conclusions clustered around themes [43]. NVivo 12 analysis software was used to create codes to group recurring words, phrases, and concepts from the interview transcripts into categories and subcategories. The coded sections were discussed iteratively within the research team (D.J.C., D.T., H.L., J.D.) to ensure agreement regarding the interpretation of the data. Emerging categories were continuously compared to new data until thematic saturation was estimated to have been achieved [44].

### 2.5. Rigour of Collected Data

Strategies to ensure the rigour of the collected qualitative data were implemented to ensure credibility, transferability, dependability, and confirmability of the findings [45]. Credibility was addressed by including member checking processes and peer debriefing in the data collection process [46]. Transferability of themes was addressed by providing generous descriptions in the collected data to ensure that conclusions could be applied to persons with similar experiences [47]. Finally, dependability and confirmability were addressed through an audit trail in NVivo 12 and interview notes to ensure that the analysis process was logical and well documented [47,48].

## 3. Results

The members of 20 families were recruited for this study. Most of the interviewed parents were mothers (n = 20), with two fathers also joining for the interview. Of the children with MDS, 17 were male (85%) and over half (55%) were aged between 11 and 20 years. Most of the families lived in the USA (n = 11), with the rest living in Australia (n = 4), Canada (n = 2), and Europe (n = 3). The participant demographics of children with MDS in this study and the interviewed parents are presented in Table 1 and Table 2, respectively.

All of the individuals in this study experienced at least one multimorbidity including seizures, gastrointestinal, and/or respiratory health problems. Three individuals had a tracheostomy and fourteen had a gastrostomy. Nearly all of the individuals were non-verbal and 80% of the individuals were unable to walk independently.

The parents described their provision of complex care for their child with MDS at home. The data were coded into three main categories: (1) complex care needs in the home, (2) highly skilled caregivers, and (3) impact on caregivers and families (see Figure 1). The categories and subcategories are presented in Table 3, Table 4 and Table 5 with sample quotes to illustrate the parent caregiver’s thoughts and experiences.

### 3.1. Complex Care Needs in the Home

The parents described their daily management regimens to support their child’s health, particularly for seizures, respiratory health, and gastrointestinal health. Surrounding these routines, the parents detailed the importance of fostering social interactions and cognitive stimulation to enhance their child’s quality of life. Sample quotes that describe home management related to the subcategories of seizures, respiratory, and gastrointestinal symptoms are presented in Table 3.

#### 3.1.1. Management of Seizures

The child’s epilepsy was managed through medication, non-medical support, surgical procedures, and the ketogenic diet. The parents used multiple medications, noting that many medications made their child drowsy and inattentive. Two out of twenty children had undergone surgical procedures including the insertion of a pulse generator for vagus nerve stimulation (VNS). One quarter of the children (n = 5) had used the ketogenic diet. Seizures necessitated additional management strategies in the immediate environment, for example, using soft materials such as pillows around their child to prevent injury during seizures.

#### 3.1.2. Management of Respiratory Health

The parents managed their child’s respiratory problems such as bronchiectasis and pneumonia through medications and other adjunct respiratory therapies. The range of respiratory medications included antibiotics, bronchodilators, and intravenous immunoglobulin treatments. Non-medication measures for managing respiratory symptoms at home included chest percussion and physical therapy, airway suctioning, high-humidity treatments, and bilevel positive airway pressure (BIPAP). Other parents managed tracheostomy or gastrostomy care as part of their regimens for maintaining respiratory health.

#### 3.1.3. Management of Gastrointestinal Health

For constipation, the main managements included the administration of oral laxatives, rectal suppositories, and enemas. The parents also used high-fibre diets and regular physical therapy to assist gut motility. Surgical procedures such as gastrostomy/gastrojejunostomy, appendicostomy/colostomy, and Nissen fundoplication had been performed for the management of feeding, constipation, and reflux, respectively. Nearly all of the parents observed improvements in their child’s gastrointestinal health following gastrostomy insertion.

#### 3.1.4. Promotion of Social Interaction, Cognitive Stimulation, and Quality of Life

The parents described the importance and methods of enhancing their child’s social interactions and cognitive functioning to improve their overall quality of life. Many parents reported positive feedback from their child following exposure to nature, sensory stimulation, and physical and social activities.

**Table 3 children-10-01202-t003:** Complex care in the home for seizures, respiratory health, and gastrointestinal health.

Seizures	
Medication	‘Over the last three years, we tried four different types of medications and none of them lessened the drops. None of them helped… She was sleepy… There was no expression. There was no giggle. There was no vocalisation.’ (Parent of 12 year old female.)
Emergency treatment	‘The problem is sometimes he will go into a seizure and not be able to come out of it. So, he’ll go into a seizure, come out of it, and two minutes later he’s back into it and that’s when he gets tired. So, he will get extremely tired and wear himself out and will stop and his breathing will get worse. So, that’s when the emergency meds come in and we try not to use them all the time… I don’t use them at random times if he does this or that. He only gets them at certain times when his seizures are really bad because he benefits off of them.’ (Parent of 17 year old male.)
Non-medical support	‘We just try to comfort him and make sure that he knows we’re there for him and we try to make sure he’s safe and doesn’t try to hurt himself… So, we just have to make sure that’s safe… that he knows we are there for him for whatever he needs.’ (Parent of 12 year old male.)
Surgical procedures	‘He has a vagus nerve stimulator that we put in that has been a big help. So, that was a good decision.’ (Parent of 27 year old male.)
Ketogenic diet	‘One of the last resorts when she was in Hopkins when she was admitted in, she was in status and basically in a medically induced coma, they said, “let’s try keto”. She went into ketosis fairly quickly and she stopped having seizures after about four weeks of being on the keto diet and she hasn’t had a seizure since that. Unfortunately, the keto diet caused other problems. Her triglycerides were through the roof, and she went into cardiac arrest. But she has not had any seizures… it’s been over a hundred days now.’ (Parent of 11 year old female.)
**Respiratory Health**	
Medication	‘He had IVIg therapy for thirteen months, and then he was out of the hospital for almost two years… When they quit the IVIg therapy, everything started to go downhill again.’ (Parent of 33 year old male.)
Non-medical support	‘We have a daily regime of her cough-assist as soon as she gets up in the morning… her puffers and nose spray and that seems to really have lessened the amount of episodes.’ (Parent of 12 year old female.)
Surgical procedures	‘He had Pierre Robin which was a small jaw and a cleft palate, and they thought that was the cause of his respiratory problems because he couldn’t swallow very well, and it was aspirating onto his lungs. So, that’s why they decided to give him the gastrostomy at the age of five weeks which saved his life.’ (Parent of 23 year old male.)
**Gastrointestinal Health**	
Constipation	‘I give her a smoothie every morning which has fruit, pureed fruit, flax seed, almond milk, orange juice, powdered greens and water. So, I really push the fluids to her, and that extra fibre and that water tends to keep things flowing. Sometimes we have had to give her a suppository or even an enema here at home, but we’ve always been able to manage it… The more active she is, the better she is… Having her in the walker, having her in the pool. That tends to get things moving.’ (Parent of 12 year old female.)
Feeding	‘Only time she aspirates since she’s had the G-tube is if we try to increase the speed of the feeding. Increasing the rate, she’ll let you know right away, she’ll start to gag. We learnt a long time ago it just wasn’t worth it because once she’d start to gag, then she would aspirate. Then we would end up in the hospital and sure enough, she would have pneumonia. So, we just run it pretty slow.’ (Parent of 11 year old female)
Surgical procedures	‘She’s had the G-tube since she was about six months old and that has worked wonderfully. We managed all her feeds through the G-tube, and she seems to handle them very well…It’s just so nice to put all her meds through it as well.’ (Parent of 11 year old female.)
**Social Interactions and Cognitive Functioning**	
Nature and sensory stimulation	‘He loves being outside and animals, the horses. Likes to be outside and wheelchair walking and enjoying nature… he gets excited when he’s outside.’ (Parent of 16 year old male.)
Physical activities	‘That was his favourite thing… swimming. He loved to swim. He loved to be outside.’ (Parent of 12 year old male.)
Social activities	‘When he’s around other people, he likes to look and watch and soak in all of the fun that’s happening. [Child] will rarely participate in those things, but he will try to tell you all about it when he gets home. He loves being around his peers.’ (Parent of 33 year old male.)

### 3.2. Highly Skilled Caregivers Delivering Complex Home-Based Medical Care

The parents attained multiple skills to provide their child’s complex care at home. Their strategies were informed through education from healthcare professionals and their lived experiences. The data were categorised into the following subcategories: prevention and early intervention, advocacy, comprehensive care, personal strengths, palliative care, and caregiver respite. Sample quotes are presented in Table 4.

#### 3.2.1. Prevention and Early Intervention

The parents reported that proactive strategies for prevention, including regular vaccinations and immunisations and early intervention, helped to improve their child’s overall health and wellbeing. Learning the early signs of their child’s symptoms of declining health and starting treatments as an immediate response were effective early intervention methods. A common strategy for illness prevention was limiting the child’s physical exposure to the general public. The parents reported that their child maintained better health with less exposure to the wider community, as occurred during the COVID-19 pandemic. However, some parents reported that their child’s mental health worsened with isolation, despite their physical health improving.

#### 3.2.2. Advocacy for Their Child’s Medical Care

The parents emphasised the importance of being affirmative and voicing their concerns to medical professionals, whereby some seemed not to appreciate the severity of their child’s condition. Advocacy was a key tactic that they needed to employ in unfavourable situations regarding their child’s medical and non-medical needs.

#### 3.2.3. Comprehensive Care

The parents elaborated on the importance of finding and establishing a medical community to support their child’s comprehensive care needs throughout their life. Many parents described challenges when their child transitioned from paediatric to adult care. Moreover, others described the importance of building a non-medical support community. Many parents described the benefits of care workers in the home to cope with the increased demands of complex care, especially overnight. Lastly, some parents reported practical organisational measures to help them coordinate care for their child.

#### 3.2.4. Personal Strengths

The parents described how they used their personal strengths as a strategy for providing complex care for their child, as well as maintaining their own mental health. Some parents reported finding solace in their faith and religion. Contrastingly, some parents preferred a more pragmatic approach.

#### 3.2.5. Palliative Care

The parents described their apprehensions and experiences regarding palliative care for their child. Some parents described hospice care as an effective strategy in ensuring a high quality of life for their child. Contrastingly, some parents were worried that the care provided in a hospice may not be to the standard that they provided at home for their child; however, for those with this experience, their concerns were resolved once they transitioned to the palliative care system.

#### 3.2.6. Caregiver Respite

The parents described the importance of regular and planned caregiver respite, in the forms of individual travel, hobbies, and even work. Furthermore, they described that taking regular breaks and incorporating non-medical routines into the day improved their own mental health, which helped them to maintain the long-term delivery of their child’s health needs.

**Table 4 children-10-01202-t004:** Strategies for home-based medical care.

Prevention and Early Intervention	
Vaccinations and immunisations	‘Luckily, he hasn’t gotten COVID. He’s had his third shot, his booster, and he did pretty well with the vaccine.’ (Parent of 19 year old male.)
Early medical intervention	‘We learnt to pick up the signs better of when she was starting to get sick and immediately jump on the pulmonary toileting and the breathing treatments… that really helped. Not made it totally stop the symptoms of respiratory infections, but it made her better faster.’ (Parent of 11 year old female.)
Health improvement through reduced physical exposure	‘Physically… yes. She did not contract any colds. Very, very healthy respiratory-wise. But mentally, she wasn’t as happy… It was almost hard to watch… She did not like the virtual schooling, so I think her mental health suffered from not being able to interact with as many people. But physically, it was great. She wasn’t around any germs, so she did really well health-wise.’ (Parent of 11 year old female.)
**Caregiver Advocacy and Affirmative Action**	
Negative encounters with medical professionals	‘Sometimes we would take him to the doctor, and they would not be as understanding that he’s not like your regular child. If he’s got a runny nose or cough or something, we got to go ahead and do something to take care of it. We can’t just wait to see if it’s going to turn into anything. Sometimes it was hard to get medical personnel to understand that was what we needed.’ (Parent of 25 year old male.)
Advocacy and courage	‘It’s hard to go into another doctor’s office and they don’t know, they just see another patient. They don’t understand him but it’s up to the parents to advocate for them and just to keep on it and say, “Well, this is what he needs, and this is who he is”. Before I had [child], I would not ask questions and I would not say this or that. I would be the one that would hide in the corner, but [child] has forced me to come out of that shell and be an advocate. That has helped me to voice my concern.’ (Parent of 17 year old male.)
**Comprehensive Care**	
Transition from paediatric to adult care	‘At the paediatric level, [child] had a hospital, he had a team of doctors there he would see, and nurses. So, they got to know him very well. But when you move into the adult world, it’s the wild west. You have doctors that have never seen these disorders. You have nurses that have never seen this disorder. Some treat them like they’re the elderly because that’s all they know.’ (Parent of 27 year old male.)
Medical community	‘I think that’s really important for parents to hear that they are their child’s expert and to find doctors and therapists and team members on the medical team that know that and respect that.’ (Parent of 4 year old male.) ‘One of the things that helped us when we moved to Florida was, we went to a group of doctors that had a wide range of specialties and they all interacted with one another with the records. That has helped a lot.’ (Parent of 25 year old male.)
Non-medical community	‘The *MECP*2 Facebook page… that’s been like a godsend… to bring all the families of children with *MECP*2 and Rett, to educate… to help… to give strength and courage to one another. I learnt so much from them.’ (Parent of 29 year old male.)
At-home medical support	‘I’d like a qualified community nurse that came in to do the overnighter or to look after him while we could go away for respite or to have a short break somewhere.’ (Parent of 23 year old male.)
Organisation	‘My husband is a big spreadsheet user… That’s how we keep track of when supplies are running out, when meds are about to run out and all of her appointments are on a shared calendar that we have. I created a file for her caregivers… how her feedings were, how her medicine works… what to do at what times.’ (Parent of 11 year old female.)
**Utilising Personal Strengths**	
Faith and religion	‘I have a good understanding of things… and a good faith. I’m largely faith-driven, so I think that helps me.’ (Parent of 17 year old male.)
Pragmatism	‘I think the Americans are a bit different to the English. They get quite religious, and God plays a big part of their child being like that. Whereas we are not very much like that… They just see things different. I suppose for them the support’s great, but we just like to be quite practical about things, not get too emotional or involve God at all because not everyone is religious.’ (Parent of 23 year old male.)
**Palliative Care**	
Hospice support	‘The best care he got is when he used to go to children’s hospice… he had really fantastic care. All the girls are qualified nurses, there’s a doctor on site all the time because it’s a hospice, and they would have [child] for three or four nights.’ (Parent of 23 year old male.)
Apprehensions	‘I think the word ‘hospice’ is scary in terms of just care and what it looks like… A lot of parents don’t want to take their child into a facility. They want to be at home to care for them. So, it’s difficult. It puts the parents in a difficult place.’ (Parent of 17 year old male.)
**Caregiver Respite**	
Travel	‘We did make a conscious decision quite early on and recognise that we both need breaks. So, we will have separate holidays… We’ll give each other a week or two a year, to go and do something away and not think about it, which is hard on the other person but it’s kind of needed.’ (Parent of 11 year old male.)
Hobbies	‘I do try to do something for myself every day, whether it’s only 20 min of stretching or if it’s painting or if it’s writing or talking to a friend… There are times I need to be reminded because I get swamped up.’ (Parent of 4 year old male.)
Work	‘I feel that my husband and I both working benefits even not just financially but also, I think for us mentally as well. Because if we had to do this 24/7, I don’t know if I’d be as good a parent to [child] if I didn’t have that break through working.’ (Parent of 11 year old female.)

### 3.3. Impact on Caregivers and Families

The parents described the significant impacts of providing complex care for their child with MDS on their own mental and physical health, family dynamics, and financial state. Sample quotes are presented in Table 5.

#### 3.3.1. Caregiver Mental Health

The daily medical care for their child with MDS affected the caregiver’s mental health and emotional wellbeing. Ongoing cycles of deterioration and improvement in their child’s health, along with recurrent hospitalisations and daily treatments and therapies, were emotionally draining for many parents. Some parents reported that they had experienced emotional trauma as a consequence. Other parents felt guilt for prioritising their child with complex medical needs over the needs of other children in the family. Some parents struggled with trusting other people to provide safe and high-quality care for their child with MDS. This could ultimately lead to isolation and strain from being solely responsible for such complex tasks. Some parents acknowledged the importance of seeking professional support to improve their mental health.

#### 3.3.2. Caregiver Physical Health

Providing complex care for children with MDS affected the physical health of caregivers. Many parents reported that daily medical tasks and routines become more challenging as their child grew older and larger and heavier in size. Furthermore, many parents reported a poor quality of life for themselves due to providing round-the-clock medical care for their child at home.

#### 3.3.3. Family Dynamics

The continuous provision of medical care in the home sometimes caused a strain on family relationships. The siblings of children with MDS had to develop independence and resilience to adjust to the various complex medical routines in the home. Furthermore, the daily provision of complex medical care for their child placed significant stress on marital relationships and dynamics.

#### 3.3.4. Financial Strain

Providing complex care put significant strain on caregiver finances. Many parents reported that they made the decision to leave their jobs to meet the demands of providing full-time care for their child at home. Other parents reported being dismissed from their workplace because they could not maintain an appropriate level of work and manage the medical demands of their child. For parents who continued working, many did not qualify for financial assistance from the government due to being classified as ‘employed’. Lastly, the family’s residential location appeared to have a significant impact on the caregiver’s financial wellbeing. The parents who lived in countries with universal healthcare, such as Australia, the UK, and Finland, described greater satisfaction with the resources available to them. The parents from the USA gave responses that appeared to vary depending on the state in which they lived. For example, some parents reported accessing government funding to provide care for their child whereas others reported having little to no funding to provide complex medical care in the home.

**Table 5 children-10-01202-t005:** Impact on caregivers.

Caregiver Mental Health	
Emotional exhaustion	‘The intensity of looking after him… you just want to lay on the ground with him and have cuddles. It’s constant therapy, if you like, that takes away all the niceties of being a parent and couple that with no positive communication from [child]. Just mentally beats you up all the time.’ (Parent of 11 year old male.)
Emotional trauma	‘I always said I have post-traumatic stress disorder because certain noises that might sound like him moving suddenly triggers me… Like if I’m doing dishes and he’s in his room and somebody else opens the door and it might sound like he fell or something… I have torn tendons in my calves from darting so I can get to him.’ (Parent of 36 year old male.)
Guilt	‘It’s a bit of guilt for me as a mom because we do have another daughter and I feel like I spend a lot of time with [child] and don’t really get to spend time with my other daughter… Even if I’m not with [child], I’m doing something for [child]… on the phone with doctors, picking up her prescriptions, managing her care.’ (Parent of 11 year old female.)
Trust issues	‘There can be some really fantastic group homes or residential settings, but no one’s ever going to love that child or that person like their family or parents do. I think that going the extra mile or getting out of bed for the everyday, that comes from love, not from duty or not from, ‘I’m getting paid to do it’, but as all parents would do for their children which is a bit different to when you are living in a residential setting, and you’ve got a rotating roster of different workers. What happens when a new worker comes into the house or none of the staff in that house know your son? So, they’re all the things that give me nightmares.’ (Parent of 17 year old male.)
Dealing with mental health	‘I’ve spoken to a psychologist at one point about carer stress. So, there is support out there if I choose to go and seek it… what he did was try to offload a bit of guilt. He did help in that respect.’ (Parent of 11 year old male.)
**Caregiver Physical Health**	
Physical strain	‘It’s strenuous physically because she is getting heavier and larger.’ (Parent of 11 year old female.)
Sleep deprivation	‘The only thing that makes it difficult is that I can’t go to bed when I’m tired… if I’m tired at ten-thirty and he’s not ready to go to bed, then I don’t get to bed. My sleep is all messed up. I don’t get enough of it… it’s hard but that’s the way it’s been all these years. I’m used to it, but just one night I would like to go to bed when I’m tired, not after. But that is not what we have here, so we just keep doing what we do.’ (Parent of 33 year old male.)
**Family Dynamics**	
Overall family stress	‘As long as we can keep him well, everything’s fine. But as soon as he starts getting ill, then it has a knock-on effect on relationships and the dynamics of the house. Everyone’s much more stressed and end up blaming each other for this and that. “It’s your turn. I’ve done his meds for the last five days and you haven’t”. It gets really awful but if we had more of a break, we probably wouldn’t be like that but it’s a lot of pressure on us.’ (Parent of 23 year old male.)
Impact on siblings	‘It’s difficult for siblings. It’s really difficult, but I find mine are really resilient because there have been so many occasions over the years where we were just so used to [child]’s poor health.’ (Parent of 8 year old female.)
Marital stress	‘With managing her care, especially like the treatments that we do in the morning and the night, because we usually do them together, we just bicker a lot you know, the way to do it or just being picky about how he does the breathing treatments versus how I do them.’ (Parent of 11 year old female.)
**Caregiver Finances**	
Difficulty of work–life balance	‘Through the thirty-three years that I cared for my son, I have lost three jobs because either a nurse didn’t show up or he was sick or whatever the case may be, and you can only miss so many days of work before they don’t want you there anymore. So, I’ve lost three jobs in my lifetime with [child].’ (Parent of 33 year old male.)
Medical expenses	‘Her roll-in shower, that was out-of-pocket. That renovation just cost us close to USD 20,000… It’s a large financial burden. Anything that is special needs or adaptive costs a lot of money and I don’t know if it’s a genuine cost or if the companies know that it’s an adapted piece or specialised.’ (Parent of 12 year old female.)
Lack of funding due to work status	‘Because both mom and dad are still together, we’re classed as ‘coping’. If I couldn’t work and we split up, I would probably get double the package than the money I get now because they seem to favour single parent families and people that don’t work as well. I’ve always worked and that’s always gone against me… If I didn’t work, I’d be much better off. But I can’t do it because my self-worth… work is also my respite I suppose.’ (Parent of 23 year old male.)
Impact of residential location	‘Up here in Scandinavia, healthcare is pretty much free. So, we have top healthcare, and we don’t need insurance to get what we want. So, it has been a good place to live with [child] because we have wheelchairs and special beds and roof lifts and everything free. All specialists and neurologists and X-rays and whatever, everything is free for us, so it’s an amazing place to live when we have a child that needs so much.’ (Parent of 16 year old male.)

## 4. Discussion

This qualitative study provides valuable insights into parent management of the complex medical care required for children with MDS, particularly for the management of epilepsy, recurrent respiratory infections, and gastrointestinal symptoms. The provision of daily complex medical care had a marked impact on the mental and physical health of parents, with a subsequent effect on family life. Such intense routines of home-based medical care are not isolated to MDS. For other neurodevelopmental disabilities, the parents of children with complex health needs have to abide by daily routines that are extremely ‘labour-intensive’, and which inhibit natural spontaneity in their lives [49,50].

### 4.1. Daily Home-Based Medical Care in the Management of MDS: Complex Routines Managed by Parents

For seizure management, many parents provided multiple anti-seizure medications to their child, although seizures often remained refractory. Moreover, their child’s attentiveness often suffered as a result of side effects [51], prompting parents to discuss with their clinician how to achieve a balance between the quantity of seizure medications and quality of life for their child [52]. Other studies have highlighted adverse associations between the use of multiple seizure medications and child quality of life in CDKL5 Deficiency Disorder [53]. The surgical insertion of a VNS enabled some reduction in the number of emergency department attendances. We acknowledge limited evidence for the effectiveness of VNS in improving seizure frequency in children with epilepsy and a need for further research in this area [54,55]. Other parents described risks and rewards in using the ketogenic diet as a treatment for their child’s seizures. Despite reductions in seizure activity with the ketogenic diet, some parents had to cease this modality due to the development of other health problems, such as pancreatitis and cardiac arrest. Although acknowledged as a treatment option for epilepsy, there is still a lack of research on the impact and risks of the ketogenic diet for seizure management [56].

The parents administered complex daily regimens to manage their child’s respiratory health. Whilst taxing, parents described how this daily routine was effective in reducing the number of respiratory infections and hospitalisations, thereby reducing burden. Furthermore, many parents also reported that gastrostomy insertion was associated with better respiratory health and a decrease in aspiration pneumonias. In contrast, a data linkage study by Jacoby et al. (2020) reported that gastrostomy for children with severe intellectual disability was associated with fewer hospitalisations overall but not reduced hospitalisations for acute lower respiratory tract infections [57].

Managing their child’s gastrointestinal health at home was a constant aspect of daily care. The parents again described the positive effects of gastrostomy but for their daily feeding and medication routines. Others mentioned that slowing the rate of gastrostomy feeding, despite prolonging their daily care routine, improved their child’s weight gain and appeared to be safer, with fewer aspiration episodes. These findings are consistent with other studies which reported that gastrostomy improved the quality of life for caregivers of children with other neurodevelopmental disorders such as cerebral palsy, as it eased medication delivery and reduced the concern about their child’s nutritional status [58,59,60,61].

In addition to medical care, parents implemented physical and social activities in their daily care routine to improve their child’s happiness and quality of life. The parents expressed how regular exposure to nature and other opportunities for sensory stimulation appeared to improve their child’s emotional wellbeing. These findings are consistent with other studies which suggest that regular involvement in community participation and social activities is associated with a better quality of life for the child [40,62].

### 4.2. Highly Skilled Caregivers Delivering Complex Home-Based Medical Care

The parents were watchful for early signs of declining respiratory health and implemented regular respiratory treatments and immunisations as preventive strategies. The parents acknowledged that by actively reducing their child’s exposure to the wider community, their child’s physical health, but not their mental health, improved. The other literature has reported that enforced isolation during the COVID-19 pandemic resulted in increased mental health issues in children, including stress, depression, and anxiety [63,64,65,66,67]. Further research exploring strategies to accommodate the child’s physical health needs without sacrificing child and family emotional health during isolation is needed, in preparation for future pandemics.

Some parents described challenges in their encounters with healthcare professionals, such as when parent concerns were dismissed. Accordingly, parents emphasised the importance of being assertive and advocating for their child in these situations, to identify and receive the best medical care. This approach is also supported in other studies that have explored the role of advocacy as a proactive response to non-supportive interactions with medical professionals, to counter the perceived lack of awareness and understanding regarding the severity and implications of a patient’s disability [68,69]. Our findings illustrate the perspectives of the parents of children with MDS regarding their complex medical care in the home. We emphasise the overarching need for improved education and awareness of this disorder for healthcare professionals and we propose that the development of clinical care guidelines in the future could in part address this need.

The parents of young adults explained that the transition from paediatric to adult care was a challenging and complex process. Many parents encountered a loss of medical resources once their child was no longer eligible to remain in the paediatric healthcare system. The difficulties of the transition between paediatric and adult healthcare are documented in other studies, suggesting a need for structured transition programs that are commenced well in advance of the time that the transition is necessary [70,71,72,73,74,75].

The parents varied in their expressed religious beliefs or took a pragmatic approach to their child’s disability, and either approach helped them to cope with the intense care demands. These findings are consistent with a study by Elnasseh et al. (2016), where the family dynamics of dementia caregivers improved by utilising empathy and open communication styles [76]. There is currently little research on the impact of caregiver personal strengths for individuals requiring complex disability care. Future studies investigating these strengths may help the families of children with neurodevelopmental disorders, such as MDS, to harness these strengths and maintain their health over the long term.

The parents had mixed feelings about the use of palliative care. A study by Kirk and Pritchard (2012) suggested that apprehensions may be due to a lack of knowledge or trust in hospice care, or even that they felt guilt about being unable to manage their child’s healthcare [77]. Targeted information regarding palliative support systems may help the parents of children with severe neurodevelopmental disorders with decisions for their child’s long-term complex care.

As we found, home-based medical care was extremely demanding on parents. Unsurprisingly, the parents of children with MDS valued having respite for themselves, even for a short period of time. Interestingly, a study by Geense et al. (2017) reported that the parents of children with a chronic kidney disease expressed little understanding of how to incorporate their own hobbies and work interests alongside the daily medical care that they provided for their child [78]. This may be due to the constant prioritisation of their child’s medical needs over their own personal needs. Our study’s findings highlight the need for more education and support for the parents of children with neurodevelopmental disabilities such as MDS on the importance of caregiver respite.

### 4.3. The Impacts of Constant Complex Care on Parents and Families

We identified some of the emotional hardships that the parents of children with MDS experience, such as mental fatigue and parental guilt. Some appreciated the benefits of seeking professional psychological support for managing their stress. Societal stigma may further contribute to poorer mental health in caregivers [79]. The parents also experienced physical stresses such as sleep deprivation. Despite a small number of studies exploring the physical toll of medical care on parents, there is little to no research regarding strategies to improve their physical health [29,30,80]. Investigating strategies to improve parent physical and mental health may not only lead to a better quality of life for parents themselves, but also to a better delivery of care for their child with complex medical needs.

The intense mental and physical stresses of home-based medical care routines also impacted other members of the household. Our research found that the siblings of children with complex health needs received less attention than parents would ordinarily choose to give, due to the intensive nature of home-based medical care [81]. Other studies have observed that the siblings of children with disabilities may feel neglect and personal burden from the intensive medical care needed by their sibling [19,82,83].

Lastly, this study reported on the financial burdens of complex care for the parents of children with MDS. The parents described how the provision of complex care was expensive due to the cost of medical treatment and equipment, exacerbated by reduced parent capacity to engage with paid employment. These findings are consistent with Baddour et al. (2021), where parents of tracheostomy-dependent children had depleted financial resources from taking loans or seeking support from external organisations to manage medical expenses [84]. Families without access to universal healthcare seemed to experience more financial challenges in their provision of care compared to families who received government assistance. We note that many families in this study lived in the USA. A study by Hiranandani (2011) suggested that there are insufficient resources in developing managed care plans for Medicaid (America’s public health insurance program) recipients with disabilities [85]. Currently, there is little research regarding the impact of other healthcare systems on the families of children with neurodevelopmental disabilities.

### 4.4. Strengths and Limitations

The open narrative approach of the interviews in our study enabled valuable insights into the lived experiences of parents who provide complex care for their children with MDS. The parent participants provided a rich set of perspectives on the complex care required for their child with MDS, as they are ideally placed to observe and understand their child’s health status and quality of life needs [86]. We acknowledge limitations. One of the limitations for this study is the small sample size. Despite achieving thematic saturation using 20 families, it is possible that having a larger proportion of families may have introduced new aspects of complex care that were not explored. The majority of parents interviewed were mothers and we collected limited information from fathers. It is important to note that the decision of which parent to participate in the interviews was chosen by the family, with no influence from the researchers. Lastly, a higher proportion of parents received government support than those who did not receive government funding, and this study may not have captured the experiences of families with even greater financial strain.

## 5. Future Directions and Conclusions

To our knowledge, this is the first study to address the home-based medical management of the comorbidities of MDS. There is a need for improved education and awareness of MDS for healthcare professionals, and the development of clinical care guidelines might address this need. There is also a need to investigate which strategies are effective for parents and siblings, to improve their mental and physical health. Lastly, future studies need to investigate caregiver perceptions of palliative care and strategies for an easier transition from paediatric to adult healthcare for those with complex neurodevelopmental disabilities. The perspectives and experiences of the parents described in this study can provide clinicians and other healthcare professionals with insights into the daily challenges of care faced by the parents of children with severe neurodevelopmental disabilities, and what they value when receiving professional care for their child.

## Figures and Tables

**Figure 1 children-10-01202-f001:**
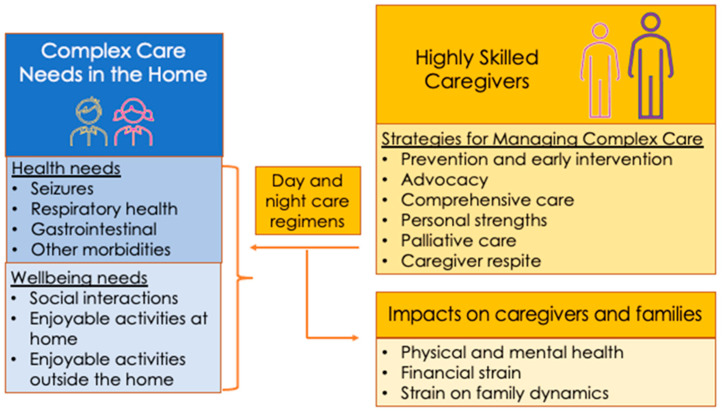
Main categories and subcategories describing how parent caregivers provide complex medical care in the home and subsequent impact on them.

**Table 1 children-10-01202-t001:** Demographics of children with MDS in this study.

Age (years)	n/N (%)	Biological Sex	n/N (%)	Comorbidities at Time of Interview	n/N (%)
1–10	1/20 (5%)	Male	17/20 (85%)	Major respiratory problems	12/20 (60%)
11–20	11/20 (55%)	Female	3/20 (15%)	Frequent seizures ^1^	17/20 (85%)
21–30	4/20 (20%)			Gastrointestinal problems	15/20 (75%)
31–40	3/20 (15%)			Tracheostomy	3/20 (15%)
Deceased	1/20 (5%)			Gastrostomy	14/20 (70%)
				Non-verbal	19/20 (95%)
				Unable to walk independently	16/20 (80%)

^1^ Two or more seizures per day.

**Table 2 children-10-01202-t002:** Demographics of interviewed parents.

Age (Years)	n/N (%)	Biological Sex	n/N (%)	Country of Residence	n/N (%)
30–39	3/22 (14%)	Male	2/22 (9%)	USA	11/20 (55%)
40–49	9/22 (41%)	Female	20/22 (91%)	Australia	4/20 (20%)
50–59	4/22 (18%)			Canada	2/20 (10%)
60–69	6/22 (27%)			Europe	3/20 (15%)

## Data Availability

The data presented in this study are available upon request from the corresponding author and subject to ethics approval.

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
