# Peer review of "How Families Manage the Complex Medical Needs of Their Children with MECP2 Duplication Syndrome"

_children, 2023, doi:10.3390/children10071202_

Round 1
Reviewer 1 Report
Congratulations. This paper deals with a topic of great scientific interest, that of rare diseases. Studies showing the experiences and perceptions of these patients are very enriching and offer a much needed perspective for this field.
The article has an appeal that keeps the reader interested throughout the reading.
From my point of view some revisions would be necessary before publication.
-Add rare disease as a keyword
-In the introduction, clarify the rationale for the article.
-In the methodology, specify in more detail what the interview was based on.
-Revise the discussion as there are paragraphs that could perfectly well appear in the introduction.
Reviewer 2 Report
Authors present a study on members of 20 patients with a methyl-CpG- binding protein 2 (MECP2) duplication syndrome (MDS) to investigate how the parents manage their child´s seizures, respiratory infections, intellectual disability and gastrointestinal symptoms and impact of it on parents. The study is nicely written and provides important insights into coping mechanisms of parents who are primary caregivers of children with a severe neurological syndrome. Drawback is that in most of the cases only one parent was interviewed (usually mother), since there is probably a difference in the point of view, dealing and impact of the child´s disease on mother and father. Results are somewhat descriptive - there are tables with senteces of parents which refer to a certain variable which was investigated; however, there was no objective assesment of stress nor use of many of the questionnaries and tests which could asses it. Therefore, this study is then a narrative survey. Also, a literature review should be expanded, and comparision with similar studies conducted, since only then we will know the true added value of this manuscript to the literature. For Introduction, I suggest to include a treatment algorithm and average life span of these patients.
For Discussion I suggest to include and comment:
Hadar-Frumer M, Ten Napel H, Yuste-Sánchez MJ, Rodríguez-Costa I. The International Classification of Functioning, Disability and Health: Accuracy in Aquatic Activities Reports among Children with Developmental Delay. Children (Basel). 2023 May 22;10(5):908. doi: 10.3390/children10050908. PMID: 37238456; PMCID: PMC10216900. Nieuwenhuijse AM, Willems DL, van Goudoever JB, Olsman E. Parent perspectives on the assessment of quality of life of their children with profound intellectual and multiple disabilities in the Netherlands. Res Dev Disabil. 2023 Jun 1;139:104536. doi: 10.1016/j.ridd.2023.104536. Epub ahead of print. PMID: 37269577.
Acceptable.
Reviewer 3 Report
Children-2462051
I read with interest the Ms. by Cherian et al. on the complex care needs at home of patients with MECP2 duplication syndrome (MDS). MDS is a rare, X-linked, neurodevelopmental disorder, which include severe intellectual disability, global developmental delay, seizures, recurrent respiratory infections, and gastrointestinal problems. Because of the presence of co-morbidities and lifelong disability of children, the patients with MDS require complex care and burden on parents/caregivers is significant. The AAs described the difficulties of the home-based medical management of the co-morbities of MDS.
A prior article described and assessed the burden on caregivers of MDS (Ak et al. 2022). The AAs should cite and discuss the results in the Ms.
The mean limitation of the study, as the AAs reported, is the qualitative study design (i.e., narrative interview).
The Ms. could be improved. Demographics data of the interview participants and MDS patients should be splitted. In my personal opinion, in order to improve the clarity of the Ms, the AAs 1) should categorize somehow the participant answers for each aspect (i.e., seizures, respiratory problems, gastrointestinal issues, etc..) and represent the identified categories graphically and describe in the text and 2) the sample quotes for each aspect could be added as supplementary data.
Round 2
Reviewer 2 Report
The authors have sufficiently responded to reviewer remarks.
Ok.
Reviewer 3 Report
I appreciated the responses to my comments. Although this is a qualitative study, the findings could of help to readership to know the complexity of this rare disease, its co-morbidities and their complex management at home.